# Histiocytic Sarcoma in a Captive Hybrid Orangutan (*Pongo* sp.): Morphological and Immunohistochemical Features

**DOI:** 10.3390/ani14060852

**Published:** 2024-03-10

**Authors:** Valentina Galietta, Niccolò Fonti, Cristiano Cocumelli, Caterina Raso, Pilar Di Cerbo, Francesca Parisi, Emanuela Bovi, Raffaella Parmigiani, Gabriele Pietrella, Antonella Cersini, Klaus G. Friedrich, Claudia Eleni

**Affiliations:** 1Istituto Zooprofilattico Sperimentale del Lazio e della Toscana “M. Aleandri”, Via Appia Nuova 1411, 00178 Roma, Italy; cristiano.cocumelli@izslt.it (C.C.); caterina.raso@izslt.it (C.R.); emanuela.bovi@izslt.it (E.B.); raffaella.parmigiani@izslt.it (R.P.); gabriele.pietrella@izslt.it (G.P.); antonella.cersini@izslt.it (A.C.); claudia.eleni@izslt.it (C.E.); 2Department of Veterinary Sciences, University of Pisa, Viale delle Piagge n.2, 56124 Pisa, Italy; niccolo.fonti@phd.unipi.it (N.F.); francesca.parisi@unipi.it (F.P.); 3Fondazione Bioparco, Viale del Giardino Zoologico 20, 00197 Roma, Italy; pilar.dicerbo@bioparco.it (P.D.C.); klaus.friedrich@bioparco.it (K.G.F.)

**Keywords:** histiocytic sarcoma (HS), non-human primates (NHPs), immunohistochemistry, neoplasia, orangutan

## Abstract

**Simple Summary:**

Histiocytic sarcoma (HS), a rare and highly aggressive cancer of histiocytic origin, has been identified across various animal species, including sporadic cases in non-human primates. This research presents the initial documented instance of disseminated HS in a captive orangutan. The clinical presentation closely resembled symptoms observed in human HS cases. A thorough examination of tissue samples revealed typical features of this cancer through histopathology, and the diagnosis was confirmed using immunohistochemistry, employing markers such as Iba-1 and HLA-DR. Notably, the lack of expression of CD163 and CD204 challenges their effectiveness as diagnostic indicators in non-human primates. This investigation contributes to our comprehension of diagnosing HS in non-human primates, emphasizing the importance of standardized markers and diagnostic procedures.

**Abstract:**

Histiocytic sarcoma (HS), an infrequent highly aggressive hematopoietic tumor, has been observed in diverse animal species, with isolated occurrences in non-human primates. This study describes the first case of disseminated HS in a 45-year-old female hybrid captive orangutan. The clinical profile mirrored symptoms seen in human HS cases, encompassing anorexia and ascites. Detailed histopathological examination demonstrated characteristic features of this tumor and immunohistochemistry, using markers such as Iba-1 and HLA-DR, confirmed the diagnosis. Significantly, the absence of CD163 and CD204 expression challenges their diagnostic utility in non-human primates. This investigation enhances our understanding of HS diagnosis in non-human primates, underscoring the necessity for standardized markers and diagnostic protocols.

## 1. Introduction

Histiocytic sarcoma (HS) is a hematopoietic tumor characterized by the proliferation of histiocytic cells. Several cases of HS have been reported in different animal species, including dogs [1], cats [2], hedgehogs [3], civets [4], and manatees [5]. In dogs, this neoplasm occurs more frequently in certain breeds, including Bernese Mountain Dogs and Flat-Coated Retrievers [1]. Conversely, it is extremely rare in human medicine, representing less than 1% of all hematolymphoid neoplasms [6]. It is also considered a rare tumor in non-human primates (NHPs), with few cases reported in the literature in various species, such as squirrel monkey, lemur, slow loris, and cynomolgus macaque [7,8,9,10,11]. Different clinical forms, ranging from localized to disseminated disease, may occur. The clinical course and prognosis of HS can vary based on several factors, including tumor location, size, presence of metastasis, and response to treatment. However, it is considered an extremely aggressive neoplasm in all species, especially in its disseminated form [6,12].

From a clinical perspective, HS often presents with nonspecific symptoms such as fever, weight loss, and anorexia, accompanied by other signs like hepatosplenomegaly, intestinal obstruction, lymphadenopathy, and pancytopenia. The pathogenesis of this disease is unclear. However, as with other hematopoietic tumors, a possible viral etiology has been hypothesized for the development of this tumor in humans and NHPs [13,14].

Histologically, histiocytic sarcoma is a round-cell tumor, with large pleomorphic cells arranged in sheets, often showing marked anisocytosis and anisocariosis [15]. Mitotic activity is high and tumors are frequently necrotic, hemophagocytosis by neoplastic cells can be a feature, and there is often a prominent mixed background inflammatory infiltrate. However, the variable histological appearance of histiocytic lesions has led to numerous inconsistencies in the terminology and diagnostic criteria for these lesions. Some types of lymphoma (anaplastic large cell lymphoma, large B-cell lymphoma, and peripheral T-cell lymphoma) are often misdiagnosed as histiocytic lesions, solely based on the morphological appearance of cells [6]. The development of immunohistochemistry has made a pivotal contribution to the differentiation of these tumors.

In 2008, the new classification of hematolymphoid tumors in human medicine defined clear diagnostic criteria using different immunohistochemical markers like CD68, CD163, Iba-1, CD4, and lysozyme [6,16]. In veterinary medicine, there is not an established unique protocol for all species. In dogs, the most used markers are CD18, Iba-1, and HLA-DR17 [17]. Only three of the six HS cases that have been reported in NHPs [7,10,11] have been immunophenotyped, with no clear evidence of appropriate markers.

The present report describes the clinical and immunophenotypic characteristics of the first case of histiocytic sarcoma in an orangutan.

## 2. Case Presentation

A 45-year-old female hybrid orangutan (*Pongo* sp.) was submitted to Istituto Zooprofilattico Sperimentale del Lazio e della Toscana for post-mortem examination. The animal lived in captivity within one of the oldest zoological gardens in Europe, the “Bioparco Zoological Garden”, in Rome (Italy). The veterinarians at the facility reported anorexia, weakness, apathy, and ascites before death. A cytological examination was performed on aspirated fluid from an abdominal centesis, revealing a population of hypersegmented neutrophilic granulocytes with pyknotic nuclei, few activated mesothelial cells, and plasma cells, consistent with a chronic inflammatory process. Blood tests revealed elevated levels of liver enzymes (GOT-AST, GPT-ALT), hyperbilirubinemia, azotemia, hyperkalemia, hypoglycemia, and hypoalbuminemia. Due to the patient’s serious and worsening clinical condition, she was euthanized.

At the time of death, the animal’s weight was 72.5 kg. The accessible mucous membranes appeared jaundiced. At necropsy, the presence of abundant serous hemorrhagic effusion and numerous widespread adhesions were appreciated in the abdominal cavity. The liver was considerably increased in volume and weight (4.3 kg; more than 2.5% of the total body weight) [18,19], pale, and with numerous multifocal well-circumscribed nodular masses with lardaceous aspect, ranging in size from a few mm to 4–5 cm in diameter (Figure 1). Similar nodules were found in the spleen and the mesentery. There was abundant tissue with a lardaceous appearance and multilobular structure on the greater curvature of the stomach and in the mesentery.

Severe mesenteric lymphadenomegaly was observed. Furthermore, marked pulmonary congestion with rare petechiae on the diaphragmatic lobes and areas of emphysema were detected.

Samples of the spleen, liver, pancreas, ovary, urinary bladder, and mesenteric lymph nodes were collected, fixed with 10% neutral buffered formalin, and processed for histopathological examination. Subsequently, 4 μm sections were subjected to hematoxylin and eosin (H&E) stain and immunohistochemical assessment. Fresh samples of liver and spleen were also collected for polymerase chain reaction (PCR) to detect Herpesviruses using a Pan-Herpesvirus molecular protocol that amplifies a conserved 215 bp–315 bp region of DNA polymerase [20].

Microscopically, the liver architecture appeared markedly altered by the presence of a multinodular, densely cellular, unencapsulated, infiltrative, and poorly demarcated neoplasm (Figure 2a). The neoplasm was characterized by the proliferation of round cells measuring 15 to 30 μm in diameter and organized in sheets, supported by a pre-existing fibrovascular stroma. Neoplastic cells were markedly pleomorphic, ranging from oval to irregular, with distinct cytoplasmic borders. The cytoplasm was moderate to abundant and eosinophilic, vacuolated, or foamy and occasionally containing erythrocytes (hemophagocytosis). Nuclei had variable shapes, ranging from round to oval and occasionally indented, with marginated chromatin containing from 1 to 2 prominent basophilic nucleoli. The degree of anisocytosis and anisokaryosis was high, with moderate karyomegaly. There were numerous multi-nucleated cells with up to six haphazardly arranged nuclei. The mitotic count was high, with 24 mitoses in 10 high-power fields. Atypical mitotic figures were encountered frequently. In the center of the neoplastic lesions, there were multifocal to coalescing areas of necrosis, and a considerable lymphocytic infiltration was predominantly noticed at the periphery of the tumor. Multifocal aggregates of neoplastic cells were also observed in the spleen, ovary, pancreas, urinary bladder, and mesenteric lymph nodes.

To determine the origin of the neoplastic cells, immunohistochemistry was performed.

Multiple sections were analyzed by the labeled streptavidin–biotin (LSAB) peroxidase method with the following antibodies: vimentin, pan-cytokeratin (AE1/AE3), CD3, CD79a, CD163, CD204, human leukocyte antigen-DR (HLA-DR), and ionized calcium binding adaptor molecule-1 (Iba1) (Table 1).

After deparaffinization in xylene and rehydration in graded alcohol according to routine procedures, heat-induced antigen retrieval in sodium citrate buffer (pH 6.0) was performed using a microwave oven for 5 min at 750 W and 10 min at 200 W. The endogenous peroxidase activity was blocked by incubating samples with the BLOXALL Blocking solution (SP-600, Vector, Burlingame, CA, USA) for 10 min and nonspecific bindings were blocked by incubation with Ultra Vision Protein Block (Thermo Fisher Scientific, Fremont, CA, USA) for 5 min. Sections were then incubated overnight at 4 °C with the primary antibodies. Serial sections of unaffected orangutan lymph nodes were used as internal positive controls. As negative controls, the primary antibodies were replaced with irrelevant and isotype-matched antibodies.

The vast majority of neoplastic cells showed intense vimentin and HLA-DR (MHC II) positivity, and a moderate-to-high expression for Iba1 (Figure 3a–c), but not for pan-cytokeratin (AE1/AE3), CD3 (Figure 3d), and CD79. The neoplastic cells also tested negative for CD163 and CD204 (Figure 3e), whose expression, on the other hand, was observed in tumor-associated macrophages (TAMs) and Kupffer cells in the liver sections. The Ki-67 proliferation index (PI) of the neoplastic cells was 31% (Figure 3f). The tumor-infiltrating lymphocytes (TILs) were mainly CD3-positive, with a scant amount of CD79a-positive B-cell lymphocytes.

## 3. Discussion

Based on the morphological and immunohistochemical findings, the tumor was diagnosed as disseminated HS. The molecular tests performed were negative for herpesvirus. Clinically, the animal presented with the same symptoms reported in cases of HS described in the human literature [6]. Recent studies have established the role of some viruses, such as retrovirus and herpesvirus in the oncogenesis of HS in NHPs [10]. In our case, we investigated only the presence of herpesvirus infection, with negative results. Therefore, further tests to investigate this aspect in the current case would be interesting.

The morphological features found in our case are similar to HS in humans and in other animal species. Nevertheless, these features, which include cellular pleomorphism, infiltrating growth, multinucleated cells, and high mitotic index, are not unique to HS. Therefore, the antibodies anti-pan-cytokeratin (AE1/AE3), CD3, and CD79a were used to exclude other tumors such as undifferentiated carcinoma or lymphoma. Histiocytic sarcoma can arise from monocyte/macrophage or dendritic cells. Defining the origin of HS tumor cells could be challenging given that some histiocytic differentiation markers are not detectable in formalin-fixed tissue sections [15]. In humans, HS is typically positive for CD163, CD68, lysozyme, and Iba-1 [6,16]. The relevance of such markers for HS in veterinary medicine is variable based on the particular species considered. For example, in dogs, CD18, HLA-DR (MHC II), and Iba-1 are the prevalent markers [15,17], whereas others are specific only to the macrophage lineage. Other markers such as CD163 and CD204 are also used; however, due to their specificity for the macrophage lineage, they often show variable expression in HS in veterinary patients, and thus are not frequently used for diagnostic purposes. [15].

To the best of the authors’ knowledge, this is the fourth paper in which data on the immunophenotype of HS in NHPs are provided [7,10,11] and the first one on orangutans. In one of these studies, performed on a squirrel monkey, Iba-1 was the only marker investigated and found to be expressed [10]. Subsequently, Sakurai et al. (2022) [11] examined CD204, CD163, HLA-DR, and Iba-1 expression in the same species. All markers were detected except for CD163. In the third case report, describing an affected cynomolgus macaque, neoplastic cells showed CD68 and lysozyme positivity [7].

In the present study, the neoplastic cells from the orangutan were positive for Iba-1, HLA-DR, and vimentin and negative for CD163 and CD204. These findings and those of Buchanan et al., 2020 [10] and Sakurai et al., 2022 [11] agree on the positivity for Iba-1 and HLA-DR. On the contrary, we did not find the same result as Sakurai et al., 2022 [11] for CD204. According to our knowledge, this marker is not typically used for the diagnosis of this tumor in human medicine, and it also shows highly variable expression in veterinary medicine [15]. Moreover, since CD204 is only expressed by monocytes/macrophages, a hypothetical dendritic cell origin of the neoplasm described in that article might explain its findings. We found no expression for CD163 and this marker was also not detected by Sakurai et al., 2022 [11]. Therefore, the relevance of CD163 for HS diagnosis in NHPs, as well as in other animal species, remains uncertain. Moreover, the Ki67 index in HS is highly variable, ranging from 5% to 50% [21], and there is no established cut-off value or prognostic value associated with its high proliferation, as seen in other tumors. 

## 4. Conclusions

Certainly, additional studies are needed to demonstrate the reliability of these histiocytic markers in HS arising in NHPs. Elucidating appropriate immunohistochemical markers will be important to both shed light on the etiopathogenesis of HS in these species and generate a standardized protocol for the diagnosis of this fatal condition.

## Figures and Tables

**Figure 1 animals-14-00852-f001:**
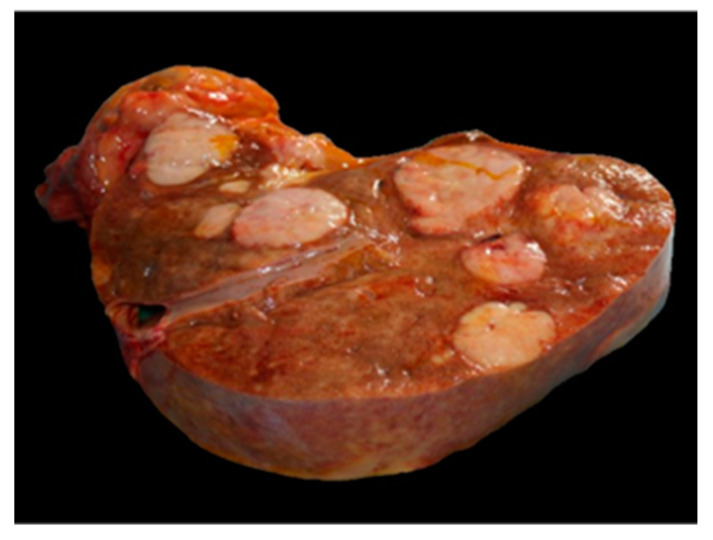
Histiocytic sarcoma, liver, orangutan. Multifocal well-circumscribed nodular masses deform liver.

**Figure 2 animals-14-00852-f002:**
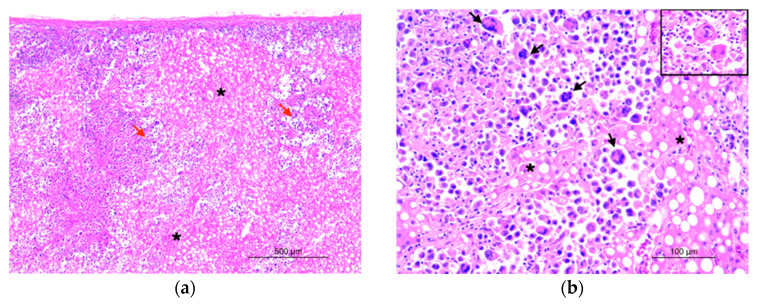
Histiocytic sarcoma, liver, orangutan. (**a**) The liver architecture (asterisks) appeared markedly altered by the proliferation of round cells organized in sheets (red arrows). Hematoxylin and eosin (HE), 5× objective. (**b**) Neoplastic cells, infiltrating hepatic parenchyma (asterisks) are round to polygonal with abundant foamy to vacuolated cytoplasm and eccentric hyperchromatic round-to-kidney-shaped nuclei. Multinucleated giant cells are also seen (black arrows). Hematoxylin and eosin (HE), 20× objective. Inset: detail of scattered multinucleated giant cells.

**Figure 3 animals-14-00852-f003:**
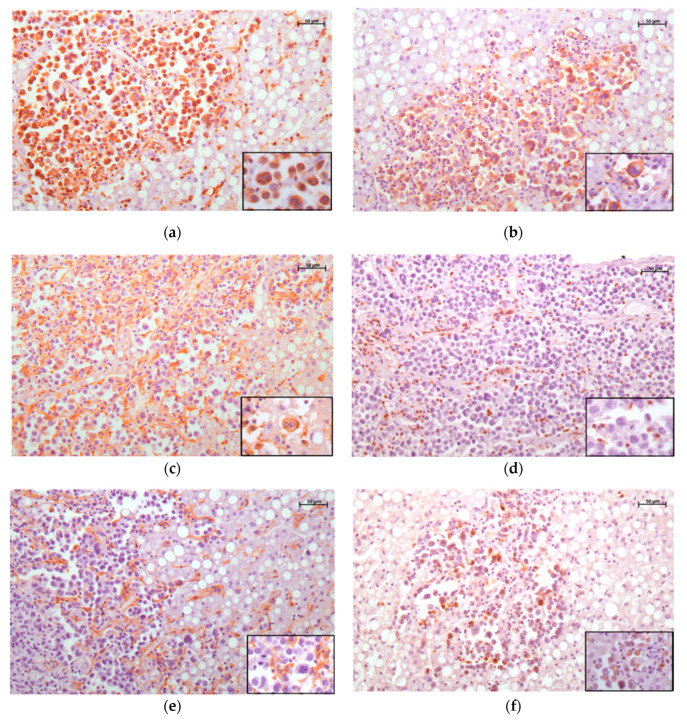
Histiocytic sarcoma, liver, orangutan. Immunohistochemistry for vimentin (**a**), HLA-DR (**b**), Iba-1 (**c**), CD3 (**d**), CD204 (**e**), and Ki67 (**f**). (**a**) Intense cytoplasmatic expression of neoplastic cells for vimentin; 20× objective (scale bar 50 μm). (**b**) Intense membranous and cytoplasmatic expression for HLA-DR; 20× objective (scale bar 50 μm). (**c**) Moderate membranous and cytoplasmatic expression of neoplastic cells, Kupffer cells, and TAMs for Iba1; 20× objective (scale bar 50 μm). (**d**) Intense CD3 cytoplasmatic expression in almost all the tumor-infiltrating lymphocytes, while the neoplastic cells show no immunoreactivity; 20× objective (scale bar 50 μm). (**e**) Lack of CD204 expression in neoplastic cells and strong membranous staining in TAMs and Kupffer cells; 20× objective (scale bar 50 μm). (**f**) Nuclear Ki67 expression with 31% Ki67 PI; 20× objective (scale bar 50 μm). In all images, inset shows neoplastic cells at higher magnification.

**Table 1 animals-14-00852-t001:** Immunohistochemical staining. The table reported primary antibodies **^a^** used and corresponding results.

Antigen ^b^	Antibody ^c^	Clone	Dilution	Source	Result
Iba-1	rabbit pAb	N/A	1:300	FUJIFILM Wako, Osaka, Japan	+
HLA-DR	mouse mAb	TAL 1B5	1:100	Novus Biologicals, Centennial, CO, USA	+
CD204	mouse mAb	SRA-E5	1:500	TransGenic Inc., Kumamoto, Japan	-
CD163	mouse mAb	AM-3K	1:75	TransGenic Inc., Kumamoto, Japan	-
CD3	rabbit pAb	N/A	1:200	Dako, Glostrup, Denmark	-
CD79a	mouse mAb	HM57	1:100	Santa Cruz, Dallas, TX, USA	-
Ki67 *	mouse mAb	MIB-1	1:100	Dako, Glostrup, Denmark	+
Vimentin	mouse mAb	V9	1:100	Dako, Glostrup, Denmark	+
pan-Cytokeratin	mouse mAb	AE1/AE3	1:100	Santa Cruz, Dallas, TX, USA	-

* Ki67: 31% positive nuclei in ten 400× fields; ^a^ the reactivity of each antibody for the different species is indicated in the website of the corresponding vendors; ^b^ Iba1 = ionized calcium binding adaptor molecule 1; HLA-DR = human leukocyte antigen-DR; ^c^ pAb = polyclonal antibody; mAb = monoclonal antibody.

## Data Availability

All data referred to in this study are available in the manuscript.

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
