# Peer review of "Histiocytic Sarcoma in a Captive Hybrid Orangutan (Pongo sp.): Morphological and Immunohistochemical Features"

_animals, 2024, doi:10.3390/ani14060852_

Round 1

Reviewer 1 Report

Comments and Suggestions for Authors

This manuscript reports the morphological and immunohistochemical features of a histiocytic neoplasm in an orangutan. Reports of neoplasia are relatively uncommon in apes and additional documentation in the current literature may help clinicians and pathologists working with these species. The manuscript presents an interesting case and is overall well-written although some additional information and revisions are needed to support and establish the importance of the authors' findings.

The clinical history, gross findings, and morphological features of the neoplastic process are well-documented and support the author’s diagnosis. The literature review, presentation, and discussion of the immunohistochemical features requires some revision, primarily Iba-1 and CD204. Specific comments with references to line numbers and tables/figures are as follows:

Line 116-117: The sentence starting “Numerous multi-nucleated cells…” is incomplete.

Line 122-123: Stating that immunohistochemistry is being used to determine the origin of the neoplastic cells, not tumor, would be more specific.  

Line 138: Suggest replacing “non-pathological” with “unaffected” or other similar phrasing.

Line 173-174 and lines 188-191: The articles cited here do not appear to discuss CD204 expression in human histiocytic sarcoma as the authors state. Has CD204 been evaluated in human histiocytic neoplasia? If other reports of CD204 in histiocytic neoplasia in humans are not found in the literature, the authors may want to take this into consideration when evaluating the absence of CD204 immunolabelling in the current case. It may also be worth noting that previous CD204 expression was documented in a new world monkey rather than an ape as in the current case. As CD240 has been documented in a NHP histiocytic neoplasm, it is important to include in regards to the diagnostic relevance of the marker, particularly if the goal is to further evaluate and develop an IHC panel to be used in various NHP species. 

Line 177 and 190: Suggest replacing “macrophagic” with macrophage.

Line 185: Is the IHC here lysosome or lysozyme?

Line 186: Suggest replacing “pathological samples” with “neoplastic cells”.

Table 1: Suggest revision of the title and addition of a legend for the abbreviations used. The spacing for the source column of the first two markers is confusing.

Figure 2c: The Iba1 immunoreactivity of the TAMs and Kupffer cells distracts from the immunoreactivity of the neoplastic cells. The authors could consider a higher magnification of the immunoreactive neoplastic cells with a smaller insert of the TAMs/Kupffer cells to serve as an internal control if journal guidelines allow for this.

Author Response

This manuscript reports the morphological and immunohistochemical features of a histiocytic neoplasm in an orangutan. Reports of neoplasia are relatively uncommon in apes and additional documentation in the current literature may help clinicians and pathologists working with these species. The manuscript presents an interesting case and is overall well-written although some additional information and revisions are needed to support and establish the importance of the authors' findings.

The clinical history, gross findings, and morphological features of the neoplastic process are well-documented and support the author’s diagnosis. The literature review, presentation, and discussion of the immunohistochemical features requires some revision, primarily Iba-1 and CD204. Specific comments with references to line numbers and tables/figures are as follows:

Line 116-117: The sentence starting “Numerous multi-nucleated cells…” is incomplete.

  • We have corrected the sentence.

Line 122-123: Stating that immunohistochemistry is being used to determine the origin of the neoplastic cells, not tumor, would be more specific. 

  • Done

Line 138: Suggest replacing “non-pathological” with “unaffected” or other similar phrasing.

  • Done

Line 173-174 and lines 188-191: The articles cited here do not appear to discuss CD204 expression in human histiocytic sarcoma as the authors state. Has CD204 been evaluated in human histiocytic neoplasia? If other reports of CD204 in histiocytic neoplasia in humans are not found in the literature, the authors may want to take this into consideration when evaluating the absence of CD204 immunolabelling in the current case. It may also be worth noting that previous CD204 expression was documented in a new world monkey rather than an ape as in the current case. As CD240 has been documented in a NHP histiocytic neoplasm, it is important to include in regards to the diagnostic relevance of the marker, particularly if the goal is to further evaluate and develop an IHC panel to be used in various NHP species. 

  • We have removed CD204 from the sentence where the markers used in human medicine for the diagnosis of histiocytic sarcoma were mentioned. Indeed, it was an error, as based on our knowledge, this marker is not typically used for this tumor in human medicine. Moreover, in veterinary medicine, especially in small animals, it is not among the most commonly used markers as it exhibits highly variable expression. For this reason, as suggested by the reviewer, we have modified the sentences related to this in the discussion.

Line 177 and 190: Suggest replacing “macrophagic” with macrophage.

  • Done

Line 185: Is the IHC here lysosome or lysozyme?

  • We have corrected this.

Line 186: Suggest replacing “pathological samples” with “neoplastic cells”.

  • Done

Table 1: Suggest revision of the title and addition of a legend for the abbreviations used. The spacing for the source column of the first two markers is confusing.

  • As suggested by the reviewer, we have modified the title of Table 1 and included a legend for the abbreviations used. Furthermore, we have revised the table according to the other recommendations provided by the reviewer.

Figure 2c: The Iba1 immunoreactivity of the TAMs and Kupffer cells distracts from the immunoreactivity of the neoplastic cells. The authors could consider a higher magnification of the immunoreactive neoplastic cells with a smaller insert of the TAMs/Kupffer cells to serve as an internal control if journal guidelines allow for this

  • We have inserted an inset with a higher magnification in Figure 2c, as suggested by the reviewer.

Reviewer 2 Report

Comments and Suggestions for Authors

I have provided minor comments to improve grammar and readability in the attached Word document.

Comments on the Quality of English Language

Only minor edits are needed to improve English grammar.

Author Response

  • We have considered all the modifications and comments provided by the reviewer on the manuscript. In particular, we report here the responses to some revisions.

Line 52 You give a very specific description of the neoplastic cells here, then in the next line, say they have a variable histologic appearance that has led to misdiagnosis. I recommend reworking this paragraph to focus on the variable histologic appearance of these challenging neoplasms.

  • We have slightly modified this sentence as suggested by the reviewer.

Line 80 Jaundice is the deposition of bilirubin in the fat leading to yellow discoloration of the tissues, not something that would be revealed on bloodwork. Do you mean hyperbilirubinemia?

  • Yes, we have corrected the term.

Figure 1 It would be better if this photomicrograph was at higher magnification or if there was an inset with a higher magnification of the neoplastic cells. It is difficult to see their morphology at this magnification. Be sure you are consistently using “multinucleated” or “multi-nucleated”.

  • As suggested by the reviewer, we have included an insert with a higher magnification (40x objective) of the neoplastic cells.

Line 117 It is typical to count 10 high-power fields or 2.37 mm2. Please redo your mitotic count to fit this convention.

  • As rightly suggested by the reviewer, we have modified the sentence by including the number of mitoses in 10 high-power fields.

Figure 2 All of these photomicrographs would benefit from being at higher magnification so that the staining of individual cells can be evaluated. I recommend choosing a different region to take the photomicrograph. One of the nodules would be better. This area doesn’t look like the other areas photographed.

  • We replaced all photomicrographs in this figure, adding in each one an inset with higher magnification.

Reviewer 3 Report

Comments and Suggestions for Authors

This article describes a case of neoplasia in a captive orangutan diagnosed as a histiocytic sarcoma and the morphologic and immunohistochemical features of this tumor in this animal. The introduction section describes the disease well with pertinent background information including both human and veterinary cases, highlighting the comparative oncology application of this case. The authors do a good job of concisely describing the clinical and gross presentation of the case as well as the histologic features and criteria used in the diagnosis.  I have a few comments.

In line 66-67, is this intended to read "HLA-DR [17,18]"? If so, is reference #17 meant to be here as it seems unrelated?

In line 81 hyperbilirubinemia would be the preferred clinicopathologic term, jaundice typically refers to the yellow discoloration resulting from hyperbilirubinemia

In Line 82, "hyperazotemia" may be an older term, "azotemia" is used more commonly

In line 87, are there established reference ranges for liver weights in this species?  If so, they should be given to demonstrate the increase in liver weight

In Fig 1b, it may be helpful to label the hepatocytes in the image

In Table 1, it would be helpful to include the species reactivity for each antibody

In line 145, is there an established threshold or prognostic value for a high Ki67 proliferation index in this tumor type?

In Fig 2, it is difficult to see the localization of the IHC staining in the images, if possible it may help to add an inset image showing a higher magnification of the IHC staining.  In Fig 2d, what is the localization of the CD3?  Also, the size of the scale bars is too small to read.

Line 172 states that some markers are not detectable in formalin fixed tissues, although IHC antibodies for some of these markers (CD11b and CD4, for example) are available for FFPE for some species (mouse, human)

Author Response

This article describes a case of neoplasia in a captive orangutan diagnosed as a histiocytic sarcoma and the morphologic and immunohistochemical features of this tumor in this animal. The introduction section describes the disease well with pertinent background information including both human and veterinary cases, highlighting the comparative oncology application of this case. The authors do a good job of concisely describing the clinical and gross presentation of the case as well as the histologic features and criteria used in the diagnosis.  I have a few comments.

In line 66-67, is this intended to read "HLA-DR [17,18]"? If so, is reference #17 meant to be here as it seems unrelated?

  • We have checked and inserted the correct reference.

In line 81 hyperbilirubinemia would be the preferred clinicopathologic term, jaundice typically refers to the yellow discoloration resulting from hyperbilirubinemia

  • As suggested by the reviewer, we have replaced the term 'jaundice' with 'hyperbilirubinemia’.

In Line 82, "hyperazotemia" may be an older term, "azotemia" is used more commonly

  • We have replaced the term 'hyperazotemia' with 'azotemia’.

In line 87, are there established reference ranges for liver weights in this species?  If so, they should be given to demonstrate the increase in liver weight

  • We thank the reviewer for the comment, but we did not find reference values for liver weight in this species in the literature. However, after reviewer’s suggestion, we decided to include two citations regarding this aspect in humans and in another non-human primate species, the macaca fascicularis. In these two studies, it is reported that the liver weight should be approximately 2.5% of the total body weight. Therefore, using these references, the weight of the orangutan in our study would be more than double what was calculated."

In Fig 1b, it may be helpful to label the hepatocytes in the image

  • We have inserted symbols to highlight the hepatocytes in this figure.

In Table 1, it would be helpful to include the species reactivity for each antibody

  • Given the large number of antibodies, it would have been challenging to include this information in Table 1. Therefore, we have decided to include the following sentence in the manuscript: ‘The reactivity of each antibody for the different species is indicated in the website of the corresponding vendors’.

In line 145, is there an established threshold or prognostic value for a high Ki67 proliferation index in this tumor type?

  • We thank the reviewer for the comment. The Ki67 index in HS is highly variable, ranging from 5% to 50% and there is no established cut-off value or prognostic value associated with its high proliferation for this tumor. So, we have added this information to the discussion and included a citation.

In Fig 2, it is difficult to see the localization of the IHC staining in the images, if possible it may help to add an inset image showing a higher magnification of the IHC staining.  In Fig 2d, what is the localization of the CD3?  Also, the size of the scale bars is too small to read.

  • We replaced all photomicrographs in this figure, adding in each one an inset with higher magnification (40x objective). Moreover we have added the scale bar to the caption and specified the cytoplasmic expression of CD3 in lymphocytes.

Line 172 states that some markers are not detectable in formalin fixed tissues, although IHC antibodies for some of these markers (CD11b and CD4, for example) are available for FFPE for some species (mouse, human).

  • As suggested by the reviewer, we have modified this sentence.

Reviewer 4 Report

Comments and Suggestions for Authors

This is a case report of a disseminated histiocytic sarcoma in a captive hybrid orangutan (Pongo sp.). Histiocytic sarcoma is a rare tumor in non-human-primates (NHP) and this is the first report in the Pongo species. This article is overall well written and implements understanding of the immunophenotype of this type of tumor in NHPs, which can aid the diagnosis. A few major edits, and mainly minor edits are suggested:

Major edits:

-          Figure 1: It would be nice to have an HE image taken at lower magnification as well (e.g., 4x objective) in addition to those already present, to better show the pattern of the neoplasm.

-          Figure 2 (d): In this picture, CD3 has a strong nuclear pattern, when it is supposed to have a cytoplasmic or membranous pattern. Also the labeled cells seem very small, even smaller than lymphocytes à Is it possible that it is labeling hematopoietic cells (like extramedullary hematopoiesis?). If confident that they are lymphocytes, do you have an explanation for the strong nuclear pattern?

Minor edits:

-          Line 37 à Reference 2 is a case report on an oral histiocytic sarcoma in a cat. Cats can have disseminated/multicentric histiocytic sarcomas as well. Referencing a book or case review on disseminated histiocytic sarcomas in cats would be a more appropriate reference then current reference 2.

-          Line 38: ‘In dogs, this neoplasm occurs more frequently in certain breeds including Bernese Mountain Dogs and Retrievers [1].’ à Reference 1 is specific to Flat-coated Retrievers (a specific breed of dog). I would recommend either specifying ‘flat-coated’ retrievers in the text, or alternatively including other references related to other retrievers (e.g., golden retriever, Labrador retriever).

-          Lines 52 through 56: Histologic features of histiocytic sarcomas. à Presence of large and multinucleated cells is also a characteristic feature of this type of tumor. Phagocytosis is not always present and can be of red blood cells (hemophagocytosis as mentioned) but also of leukocytes and tumor cells. When hemophagocytosis is prevalent, it would lead to diagnosis of a specific type of histiocytic sarcomas, i.e., hemophagocytic histiocytic sarcoma. I would recommend rewording this short paragraph. This information can be extracted from reference 15 (Tumors in domestic animals, 5th edition).

-          All figures: 20x. à I understand that 20x is the objective. I would recommend specifying ‘objective’ in all figures’ captions (e.g., 20x objective), so that it doesn’t get confused with the overall magnification (objective + ocular à which would be 200x using a 20x objective and 10x ocular).

-          Line 100: ‘fixed with 10% neutral buffered formalin’ à Please specify for how long, since a prolonged fixation time can impair immunohistochemical analysis results.

-          Line 103: ‘Samples of liver and spleen’. à I assume these were fresh samples (not fixed). Please specify for clarity.

-          Line 107: ‘the liver structure’ à I would recommend ‘architecture’ instead of structure.

-          Line 110: ‘organized in sheets’ à For completeness, I would suggest including the type and amounts of stroma too.

-          Line 116: ‘Numerous multinucleated cells with up to six haphazardly arranged nuclei’ à This sentence is incomplete (no verb).

-          Line 117: ‘with 1 to 3 mitoses in one high-power field (hpf; 0.237 mm2) à 0.237 square mm would be if you used a 40x objective and a 10x F22 ocular (= 400x field). If a different ocular (not F22) was used, then the field size should be recalculated (will not be 0.237 mm2). Could you also add how many mitoses were counted in ten 400x fields (2.37 mm2)?

-          Table 1: ‘the colons reported the primary antibody used and corresponding results’ à Was colons supposed to be the columns?

-          Table 1: Please add a legend including pAb = polyclonal antibody, etc.

-          Table 1: Ki67: 31% positive nuclei in 10 HPF à ten 400x fields, or 2.37 square mm would also work instead of 10 HPF.

-          Line 141: ‘a moderate to intense positivity’ à Is it in terms of intensity of immunoreaction or in terms of number/% of cells that are positive? I would recommend giving an idea of both (e.g., all neoplastic cells/the vast majority of neoplastic cells/50% of the neoplastic cells had immunoreactivity with moderate to high intensity for…).

-          Line 159: ‘The virological tests performed’ à I would recommend replacing ‘virological’ with ‘molecular’. Virological may infer viral isolation rather than PCR testing.

-          Discussion: Ki67 PI was reported but not discussed. Comparison with histiocytic sarcomas from other species, or with other tumors?

Comments on the Quality of English Language

This article is overall well written and only requires minor editing of English language.

Author Response

This is a case report of a disseminated histiocytic sarcoma in a captive hybrid orangutan (Pongo sp.). Histiocytic sarcoma is a rare tumor in non-human-primates (NHP) and this is the first report in the Pongo species. This article is overall well written and implements understanding of the immunophenotype of this type of tumor in NHPs, which can aid the diagnosis. A few major edits, and mainly minor edits are suggested:

Major edits:

Figure 1: It would be nice to have an HE image taken at lower magnification as well (e.g., 4x objective) in addition to those already present, to better show the pattern of the neoplasm.

  • As suggested by the reviewer, we have added a figure in hematoxylin and eosin taken at lower magnification (5x objective) to better illustrate the tumor growth pattern. So, we have modified the figures, separating the photo of macroscopic liver lesions (Figure 1) from the histological ones (Figure 2a and 2b).

Figure 2 (d): In this picture, CD3 has a strong nuclear pattern, when it is supposed to have a cytoplasmic or membranous pattern. Also the labeled cells seem very small, even smaller than lymphocytes Is it possible that it is labeling hematopoietic cells (like extramedullary hematopoiesis?). If confident that they are lymphocytes, do you have an explanation for the strong nuclear pattern?

  • We agree with the reviewer that in the Figure 2d, the pattern of CD3 appeared to be nuclear. However, we believe this was due to a very intense reaction and the low amount of cytoplasm in these cells. Therefore, we have decided to replace the previous image with a new one that better shows the cytoplasmic localization. The new image also includes an inset, at a higher magnification (40x objective), which allows for a better view of this detail.

Minor edits:

Line 37 Reference 2 is a case report on an oral histiocytic sarcoma in a cat. Cats can have disseminated/multicentric histiocytic sarcomas as well. Referencing a book or case review on disseminated histiocytic sarcomas in cats would be a more appropriate reference then current reference 2.

  • As recommended by the reviewer, we have replaced reference 2 with a more suitable one.

Line 38: ‘In dogs, this neoplasm occurs more frequently in certain breeds including Bernese Mountain Dogs and Retrievers [1].’ Reference 1 is specific to Flat-coated Retrievers (a specific breed of dog). I would recommend either specifying ‘flat-coated’ retrievers in the text, or alternatively including other references related to other retrievers (e.g., golden retriever, Labrador retriever)

  • We have corrected the name of the canine breed, as rightly suggested by the reviewer.

Lines 52 through 56: Histologic features of histiocytic sarcomas. Presence of large and multinucleated cells is also a characteristic feature of this type of tumor. Phagocytosis is not always present and can be of red blood cells (hemophagocytosis as mentioned) but also of leukocytes and tumor cells. When hemophagocytosis is prevalent, it would lead to diagnosis of a specific type of histiocytic sarcomas, i.e., hemophagocytic histiocytic sarcoma. I would recommend rewording this short paragraph. This information can be extracted from reference 15 (Tumors in domestic animals, 5th edition).

  • The degree of hemophagocytosis was not predominant enough to suggest a hemophagocytic form of HS. Therefore, upon the reviewer's suggestion, we have decided to modify this sentence.

All figures: 20x. I understand that 20x is the objective. I would recommend specifying ‘objective’ in all figures’ captions (e.g., 20x objective), so that it doesn’t get confused with the overall magnification (objective + ocular which would be 200x using a 20x objective and 10x ocular).

  • We have modified the caption by adding the term 'objective'.

Line 100: ‘fixed with 10% neutral buffered formalin’ Please specify for how long, since a prolonged fixation time can impair immunohistochemical analysis results.

  • As suggested by the reviewer, we have included the fixation time in the manuscript.

Line 103: ‘Samples of liver and spleen’. I assume these were fresh samples (not fixed). Please specify for clarity

  • Done

Line 107: ‘the liver structure’ I would recommend ‘architecture’ instead of structure.

  • Done

Line 110: ‘organized in sheets’ For completeness, I would suggest including the type and amounts of stroma too.

  • As suggested by the reviewer, we have added the description of the stroma in the manuscript.

Line 116: ‘Numerous multinucleated cells with up to six haphazardly arranged nuclei’ This sentence is incomplete (no verb).

  • We have corrected this sentence.

Line 117: ‘with 1 to 3 mitoses in one high-power field (hpf; 0.237 mm2) 0.237 square mm would be if you used a 40x objective and a 10x F22 ocular (= 400x field). If a different ocular (not F22) was used, then the field size should be recalculated (will not be 0.237 mm2). Could you also add how many mitoses were counted in ten 400x fields (2.37 mm2)?

We have corrected the sentence, inserting the number of mitoses in 10 high-power fields.

Table 1: ‘the colons reported the primary antibody used and corresponding results’. Was colons supposed to be the columns?

  • We have modified the title of this table, because we realized that the description was not very clear.

Table 1: Please add a legend including pAb = polyclonal antibody, etc.

  • Done

Table 1: Ki67: 31% positive nuclei in 10 HPF ten 400x fields, or 2.37 square mm would also work instead of 10 HPF.

  • Done

Line 141: ‘a moderate to intense positivity’.Is it in terms of intensity of immunoreaction or in terms of number/% of cells that are positive? I would recommend giving an idea of both (e.g., all neoplastic cells/the vast majority of neoplastic cells/50% of the neoplastic cells had immunoreactivity with moderate to high intensity for…).

  • As suggested by the reviewer, we have replaced this sentence with the following: ‘The vast majority of neoplastic cells showed intense vimentin and HLA-DR (MHC II) positivity, and a moderate to high expression for Iba1’.

Line 159: ‘The virological tests performed’. I would recommend replacing ‘virological’ with ‘molecular’. Virological may infer viral isolation rather than PCR testing.

  • Done

Discussion: Ki67 PI was reported but not discussed. Comparison with histiocytic sarcomas from other species, or with other tumors?

  • We thank the reviewer for the comment. Based on our knowledge, there is limited information available on the value of Ki67 in this tumor, both in human and veterinary medicine. The Ki67 index in HS is highly variable, ranging from 5% to 50%, and there is no established cut-off value or prognostic value associated with its high proliferation, as seen in other tumors. As suggested by the reviewer, we have added this information to the discussion and included a citation about this.

Round 2

Reviewer 1 Report

Comments and Suggestions for Authors

Thank you for correcting the information regarding CD204 and the formatting in Table 1. In Table 1, is 'colons' meant to be 'columns'?

Figure 2 and 3: I would recommend moderately increasing the size of the insets and adding a scale bar to each. All images in Figure 3 are also dark and could benefit from white balance and improved focus.

Comments on the Quality of English Language

Overall the language is good with minor errors throughout the manuscript.